# *Helicobacter pylori* Stress-Response: Definition of the HrcA Regulon

**DOI:** 10.3390/microorganisms7100436

**Published:** 2019-10-11

**Authors:** Davide Roncarati, Eva Pinatel, Elisabetta Fiore, Clelia Peano, Stefany Loibman, Vincenzo Scarlato

**Affiliations:** 1Department of Pharmacy and Biotechnology (FaBiT), University of Bologna, 40126 Bologna, Italy; davide.roncarati@unibo.it (D.R.); elisabetta.fiore@uni-wuerzburg.de (E.F.); sloibman@gmail.com (S.L.); 2Institute of Biomedical Technologies, National Research Council, 20090 Milan, Italy; eva.pinatel@itb.cnr.it; 3Institute of Genetic and Biomedical Research, UoS Milan, National Research Council, Rozzano, 20089 Milan, Italy; clelia.peano@humanitasresearch.it; 4Genomic Unit, Humanitas Clinical and Research Center, Rozzano, 20089 Milan, Italy

**Keywords:** heat-shock response, stress conditions, HrcA repressor, *Helicobacter pylori*, RNA-sequencing, transcriptome

## Abstract

Bacteria respond to different environmental stresses by reprogramming the transcription of specific genes whose proper expression is critical for their survival. In this regard, the heat-shock response, a widespread protective mechanism, triggers a sudden increase in the cellular concentration of different proteins, including molecular chaperones and proteases, to preserve protein folding and maintain cellular homeostasis. In the medically important gastric pathogen *Helicobacter pylori* the regulation of the principal heat-shock genes is under the transcriptional control of two repressor proteins named HspR and HrcA. To define the HrcA regulon, we carried out whole transcriptome analysis through RNA-sequencing, comparing the transcriptome of the *H. pylori* G27 wild type strain to that of the isogenic *hrcA*-knockout strain. Overall, differential gene expression analysis outlined 49 genes to be deregulated upon *hrcA* gene inactivation. Interestingly, besides controlling the transcription of genes coding for molecular chaperones and stress-related mediators, HrcA is involved in regulating the expression of proteins whose function is linked to several cellular processes crucial for bacterial survival and virulence. These include cell motility, membrane transporters, Lipopolysaccharide modifiers and adhesins. The role of HrcA as a central regulator of *H. pylori* transcriptome, as well as its interconnections with the HspR regulon are here analyzed and discussed. As the HrcA protein acts as a pleiotropic regulator, influencing the expression of several stress-unrelated genes, it may be considered a promising target for the design of new antimicrobial strategies.

## 1. Introduction

*Helicobacter pylori* is a gastric human pathogen, whose medical importance is due to its ability to promote the development of several gastric diseases, including peptic ulcer and malignant gastric cancer [1,2]. A persistent infection of *H. pylori* relies on the expression of several virulence factors and of some heat-shock proteins (HSPs), which seem to have additional roles during the infectious process [3]. Accordingly, *H. pylori* finely controls the expression of HSPs by implementing complex strategies upon perception of changing environmental inputs. HSPs regulation in *H. pylori* is negative, being managed by two heat-shock transcriptional regulators, named HspR and HrcA. These two regulators, highly widespread among bacteria as negative regulators of heat-shock response [4], bind to some heat-shock promoters to tightly modulate the expression level of HSPs and, hence, assuring their correct amount inside the bacterial cell. In particular, previous studies by our group showed that HspR alone or in combination with HrcA repress the transcription of the three multicistronic operons (*groES-groEL*, *hrcA-grpE-dnaK*, and *cbpA-hspR-rarA*), coding for the almost complete chaperone repertoire of *H. pylori* [5,6,7,8]. In fact, HspR alone is able to control its own transcription by binding to the promoter upstream the *cbpA* gene, while the regulation of *groES-groEL* and *hrcA-grpE-dnaK* operons’ transcription is dependent on both HrcA and HspR proteins. Considering these regulatory interactions, it appears that the HrcA regulon is totally enclosed inside the regulon of the master repressor HspR, a rare regulatory scheme that resembles the heat-shock network described in *Staphylococcus aureus* [4].

Although the molecular details of HspR and HrcA regulation on chaperones’ promoters have been extensively studied (recently reviewed in [3]), genome-wide studies addressed at the dissection of the possible involvement of these repressors in controlling other targets, beyond the core heat-shock genes, and other cellular processes have been overlooked until recent years. In a recent study by our group aimed at the further definition of HspR role in *H. pylori*, ChIP-sequencing and RNA-sequencing approaches have been used to identify the HspR regulon and to discriminate between direct regulation and indirect effects. These data showed that HspR participates in the regulation of tens of genes coding for proteins involved in different cellular crucial functions and is not necessarily connected to stress response. In addition, ChIP-seq data showed that HspR binds to a limited number of genomic sites, supporting the hypothesis that the contribution of HspR to the heat-shock response regulation is largely indirect [9].

In this study, we expanded our analysis to the identification of the HrcA regulon by carrying out a whole transcriptome profiling to globally identify the genes affected by HrcA inactivation. Results were compared to the available heat-shock regulon and to the HspR regulon. This analysis revealed that, besides controlling some major stress-related chaperones, the HrcA protein is engaged in the regulation of different cellular processes including cell motility, transport of small molecules across the membrane and host-pathogen interaction.

## 2. Materials and Methods 

### 2.1. Bacterial Strains and Growth Conditions

Bacterial cells used in this study (*H. pylori* G27 wild type strain [10] and the isogenic (*hrcA::km*) *hrcA* deletion mutant [6]) were recovered from frozen glycerol stocks on modified Brucella broth agar plates containing 5% fetal calf serum (FCS), in a controlled atmosphere (9% CO_2_–91% air) at 37 °C in a water-jacketed incubator (Thermo Forma Scientific, Waltham, MA, USA) for two–three days. Liquid cultures were carried out at 37 °C in modified Brucella broth supplemented with 5% FCS, with gentle agitation (130 rpm) for two days, diluted in fresh medium and grown to mid-exponential phase (about 6 h).

### 2.2. RNA Extraction

*H. pylori* cells were liquid-grown as detailed above until the mid-exponential phase (OD = 0.6–0.7). Then, ten ml of culture was mixed with 1.25 mL of ice-cold EtOH-phenol stop solution (5% acid phenol-95% EtOH) to preserve RNA integrity. Cells were harvested and then lysed to extract total RNA with TRI-reagent (Sigma-Aldrich, St Louis, MO, USA), following the manufacturer’s instructions. Prior to use, RNA samples were analyzed through agarose gel electrophoresis to control RNA purity and integrity.

### 2.3. RNA-Sequencing: Library Preparation, Sequencing and Analysis

For each condition and replica, rRNAs were depleted by using the RiboZero Gram negative kit (Epicentre, Illumina, Madison, WI, USA) starting from 1 μg of total RNA and strand specific RNA-seq libraries were prepared by using the ScriptSeq^TM^ v2 RNAseq library preparation kit (Epicentre, Illumina), from 50 ng of previously rRNA-depleted RNA. Then, 85 bp reads were produced on a GAIIX Illumina sequencer obtaining a minimum of 8 Million reads per sample and Bowtie 2 (v2.2.6) [11] was used to align raw reads to *H. pylori* G27 genome, obtaining between 89% and 94.5% of mapped reads. A modified version *H. pylori* G27 annotation based on RefSeq GCF_000021165.1, BEDTools (v2.20.1) [12] and SAMtools (v0.1.19) [13] were used to verify the library preparation and sequencing performances and to produce strand specific gene level counts. Ribosomal RNA depletion produced a reduction of ribosomal reads to less than 6% of the total mapping, 99% of the annotated transcripts were covered by at least one strand specific read and a minimum of 100 reads were counted on 90% of them. The R package DESeq2 (v1.4.5) [14] was then used to normalize the counts and to identify differentially expressed genes (DEGs) showing BH (Benjamini-Hochberg) adjusted *p*-value (padj) lower than 0.01 and log2 fold changes (log2FC) > |1|. To evaluate functional enrichments in the DEGs lists, we retrieved their clusters of orthologous groups (COG) functional classes from the NCBI CDD database [15]. Please refer to Pepe et al. [9] for further details. Raw data are publicly available at the Sequence Reads Archive under accession number BioProject PRJNA421261 and PRJNAXXXXX.

### 2.4. Quantitative Real Time PCR (qRT-PCR) Analysis

cDNA synthesis and qRT-PCR analysis were carried out as previously reported [16]. Briefly, to eliminate genomic DNA contamination, RNA samples were digested with 1 Unit of DNase I for 45 min at 37 °C. Then, 1 μg of purified RNA was reverse transcribed using 50 ng of random primers (Invitrogen, Carlsbad, CA, USA), dNTPs (1 mM each), AMV-Reverse Transcriptase (Promega, Madison, WI, USA) and incubating the reaction at 37 °C for 60 min. For real time PCR assays, 2 μL of the ten-fold diluted cDNA samples were mixed with 5 μL of 2× qPCRBIO SyGreen Mix LO-ROX (PCR BIOSYSTEMS) and specific primers mapping in the coding sequence of the genes of interest (Appendix A) at 400 nM concentration in a final reaction volume of 10 μL. qRT-PCR was carried out as follows: Initial denaturation at 95 °C for 2 min, then 40 cycles consisting of a denaturation step at 95 °C for 5 s followed by 30 s at 60 °C. For each qRT-PCR assay, a melting curve was included at the end of the amplification cycles to check for the specificity of the reaction. Data analysis was done by applying the ΔΔCt method, in which the 16S rRNA gene was used as internal reference for data normalization. Real Time PCR of 16S rRNA on cDNA samples from *H. pylori* G27 wild type and *ΔhrcA* cells gave overlapping amplification plots, indicating that the expression of 16S rRNA was unaffected by *hrcA* deletion.

### 2.5. Immunoblot Analysis

Immunoblotting was performed as previously described [17]. Briefly, equal amounts (10 μg) of total protein extracted from *H. pylori* G27 wild type and *hrcA*-mutant cells were separated through SDS-PAGE and blotted onto a nylon (PVDF) membrane by means of a wet-transfer apparatus (BioRad). Following 1 h incubation at room temperature in blocking buffer (1× PBS containing 0.05% Tween 20 and 5% low-fat milk), the membrane was stained with a 1:5000 dilution of α-BabA and α-HP1043 antibodies for 16 h at 4 °C in blocking buffer. 

After washing the membrane in PBST (1× PBS; 0.05% Tween-20), it was incubated with 1:5000-diluted horseradish peroxidase-conjugated α-rabbit antibody for 1 h at 25 °C (Thermo Fisher Scientific, Waltham, MA, USA). Following an additional washing step in PBST, the membrane was developed by pouring on it a solution of 1.25 mM luminol containing 0.015% H_2_O_2_ and 0.068 mM p-coumaric acid.

## 3. Results

### 3.1. RNA–Seq Analysis Reveals the HrcA-Dependent Transcriptome

In order to detect HrcA-dependent changes in gene expression, we carried out whole transcriptome RNA sequencing analysis of the *H. pylori* G27 wild type and of the isogenic *hrcA* deletion mutant (*hrcA::km*) strains. Data analysis revealed that 49 genes were differentially expressed (log2 fold change >1 or <−1) in the *hrcA-*mutant (Figure 1, Table 1 and Appendix A), and among them, 16 were up-regulated and 33 were down-regulated. 

As HrcA represses the transcription of the *groES* and *groEL* genes, their transcripts were accordingly found in the up-regulated list of genes of our analysis (Table 1 and Appendix A). On the contrary, the transcript amounts of the *grpE* and *dnaK* genes (HPG27_RS00570–HPG27_RS00575), belonging to a well-known HrcA target operon, were unaffected by *hrcA* mutation, likely because of a polar effect of the inactivation of the first gene of the operon (*hrcA*). 

To get a general view of HrcA regulatory function, a functional enrichment analysis was performed, and protein-coding sequences were classified using Clusters of Orthologous Groups of Proteins (COGs) Database [15]. According to this analysis, among the up-regulated genes we noticed some functional groups of particular interest (Figure 2A). For example, we identified genes (HPG27_RS03705 and HPG27_RS08000) belonging to the “coenzyme transport and metabolism” category and a group of genes (*dppB*, *dppC*, *dppD*, *dppF*) that constitute a multicistronic operon (according to DOOR, the Database of prOkaryotic OpeRons) coding for components of ABC transporters and, hence, related to “inorganic ion transport and metabolism”. Moreover, of particular interest are some genes coding for lipopolysaccharide (LPS) biosynthesis proteins (HPG27_RS01045, HPG27_RS03015), belonging to the “cell wall/membrane/envelope biogenesis” functional group (Figure 2A, Table 1 and Appendix A).

The analysis of transcripts downregulated in the *hrcA*-mutant shows that the predominant functional category is represented by “cell motility” (Figure 2A), comprising several genes encoding subunits of the flagellar machinery and proteins involved in the proper assembly of the bacterial flagella (*flaB*, *flgL*, *flaA*, *fliD*, *fliS*, *flgE*, *fliK*, *flgK*, *fliW*). This finding is in agreement with previous observations that a *H. pylori hrcA* deletion mutant shows motility defects when inoculated into semisolid agar plates [7]. In addition, among the down-regulated genes in the *hrcA*-mutant background, we found four genes coding for transposases and, hence, belonging to the functional category “mobilome: Prophages and transposon” (Figure 2A).

### 3.2. Comparison of the HrcA regulon with the HspR and Heat-Shock Regulons

Subsequently, the above results were compared to the previously identified deregulated genes upon exposure of the *H. pylori* G27 wild type strain to heat-shock at 42 °C and to the genes deregulated in the *hspR* deletion mutant [9]. The Venn diagrams reported in Figure 2 represent the number of genes coherently up- (Figure 2B) or down-regulated (Figure 2C) in the different conditions described above. Only three genes out of 16 were up-regulated by *hrcA* deletion and by heat-shock treatment and two of them (*groES* and *groEL*) were also up-regulated in the *hspR*-mutant, while one (a hypothetical adhesin coding gene, *sabB*) was deregulated in the *hrcA*-mutant and upon heat-shock treatment (Figure 2B). By contrast, four genes (HPG27_RS00625, *dppC*, *fecA2*, HPG27_RS08000), not responsive to heat-shock, were up-regulated in both the *hrcA* and *hspR* deletion mutants. 

Of the 33 down-regulated genes in the *hrcA*-mutant, none of them were down-regulated also upon heat treatment (Figure 2C), while six genes (HPG27_RS00240, HPG27_RS02575, HPG27_RS03550, HPG27_RS04645, HPG27_RS04725, HPG27_RS04735) of the mobilome/replication functional categories were down-regulated in both *hrcA* and *hspR* mutants (Figure 2C). 

Surprisingly, we identified 10 genes that were repressed in the *hrcA*-mutant and induced by heat-shock treatment, thus showing opposite behaviour. Moreover, among these genes, five were also down-regulated in the *hspR*-deletion mutant, hence showing a complex regulatory pattern (Appendix A). These latter include most of the genes mentioned above to be *hrcA*- and *hspR*-downregulated (HPG27_RS00240, HPG27_RS02575, HPG27_RS03550, HPG27_RS04645, HPG27_RS04735).

Intriguingly, our transcriptome study shows that only a small fraction of the genes belonging to the HrcA regulon are also heat-responsive (that is, deregulated in the wild type strain subjected to heat-shock), while the major part of them appears to be heat-stress independent. 

### 3.3. RNA-Seq Data Validation

To independently validate the RNA-seq results, RNA transcripts abundance of 14 newly identified HrcA-regulated genes and of *groEL* (positive control) were measured by quantitative real-time PCR, using specific primers for each sequence of interest. Specifically, as highlighted in grey in Table 1, the transcript levels of five up-regulated genes and of 10 down-regulated genes were compared in the wild type and in the *hrcA*-mutant strains. For data analysis, *ΔhrcA* vs wt fold-change ratios of ≥ 2.0 or ≤ 0.5 were considered as validated up- and down-regulations, respectively.

For all the genes included in this analysis, we confirmed a significant *hrcA*-dependent de-regulation in the mutant strain compared to the wild type (Figure 3A,B). To further support our findings, we investigated the expression of the outer membrane protein BabA in the wild type and *hrcA-*mutant strain through immunoblotting on wild type and *ΔhrcA* total protein extracts, using a specific α-BabA antibody. According to our RNA-seq results, the BabA encoding gene (HPG27_RS01595, sometimes referred as *omp28*) appears to be significantly down-regulated in the *hrcA*-mutant strain (log2 fold change −1.21, Appendix A). As shown in Figure 3C, immunoblot analysis on three different wild type and *ΔhrcA* independent biological replicates showed that BabA expression was lower in the *hrcA*-knockout strain than in the wild type, confirming HrcA involvement in BabA regulation and further supporting our whole transcriptomic analysis.

## 4. Discussion

Whole transcriptome analysis through RNA-sequencing of *H. pylori hrcA*-mutant strain revealed that this heat-shock repressor is directly or indirectly involved in the regulation of 49 genes (Figure 1) with disparate functions. Considering that *H. pylori* possesses two heat-shock transcriptional repressors, HrcA and HspR, which are responsive to temperature stress [19,20], we compared the transcriptome of the *H. pylori* G27 *hrcA*-mutant strain to the parental strain submitted to heat-shock and to the *H. pylori* G27 *hspR*-mutant strain (dataset derived from [9]). Strikingly, the vast majority of genes belonging to the HrcA regulon is not coherently induced by heat-shock, at least in the experimental conditions used, being limited to only three genes up-regulated in both conditions (Figure 2B). Another intriguing observation emerging from this analysis is that the overlap between the HrcA and the HspR regulons is restricted to a limited number of genes, mainly represented by the already known HrcA and HspR co-regulated HSP genes (Figure 2C). The global picture deriving from this study describes the *H. pylori* HrcA protein as a pleiotropic regulator with a specific (i.e. HspR-independent) regulon, influencing the expression of several stress-unrelated genes (Figure 2A,B,C). In order to define HrcA regulon, genome-wide expression approaches have been applied in other bacterial species, even though such studies are still limited for this repressor. What generally emerges is that many genes coding for proteins involved in diverse cellular processes are dependent on HrcA regulation. For instance, it has been shown that in the food-borne pathogen *Listeria monocytogenes*, several core cellular functions, including chromosome replication, protein synthesis and stress response, are affected by HrcA [21]. In addition, *hrcA* gene inactivation provokes a similar pleiotropic gene deregulation also in the Gram-positive lactic acid bacterium *Lactobacillus plantarum* [22].

One limitation of our study is that several attempts to discriminate between the HrcA direct or indirect contribution in the regulation of the genes belonging to its regulon through genome-wide approaches, such as ChIP-seq assay, were unsuccessful. On the contrary, this kind of analysis has been set up for the other *H. pylori* heat-shock regulator HspR, leading to the identification of its genomic binding sites, constituted by few promoters driving the transcription of genes coding for chaperones [9]. Further efforts are needed to identify HrcA target promoters, thereby defining the extension of the HrcA direct regulon.

A crucial aspect of HrcA regulation, supported by the data here presented, is the link between this heat-shock repressor and motility of the bacterium. As in many bacterial pathogens, *H. pylori* relies on the coordinated and hierarchical expression of flagellar genes to successfully colonize the host and to find its proper niche. RNA-sequencing analysis highlights the *hrcA*-dependent down-regulation of many transcripts coding for components of the *H. pylori* unipolar flagella (*flaA*, *flaB*, *fliD*, *flgK*, *flgL* and *flgE*) and for proteins involved in the biosynthesis of the motility apparatus (*fliK*, *fliW* and *fliS*) (Figure 4, Table 1 and Appendix A).

This finding supports a previous observation that the *H. pylori hrcA*-mutant strain shows a non-motile phenotype when assayed on soft-agar plates [7]. In addition, an intimate relationship between motility and heat-shock gene regulation has been proposed in *Campylobacter jejuni*, a close phylogenetic relative of *H. pylori* [23]. Furthermore, a recent work proposes that the *H. pylori* flagella could play a pivotal structural role during biofilm formation [24]. In this latter study, comparative transcriptome profiling between *H. pylori* planktonic and biofilm cells highlighted differential expression of several flagellar genes between the two different modes of growth. Specifically, several flagellar components (such as FlaB and FlaG), other structural components (FlgE flagellar hook protein, FlgB rod protein, FlgK-FlgL hook-filament junction proteins) and flagellar regulatory players (as the hook length control protein FliK) were biofilm up-regulated. A similar correlation between biofilm formation and expression of flagellar genes has also been described in the model organism *Escherichia coli* [25]. Interestingly, in the same study, Hathroubi and colleagues observed that *hrcA* gene is up-regulated in the biofilm transcriptome, suggesting an intimate link among HrcA, biofilm formation and motility [24].

Of note is the observation that among the up-regulated genes, most of them code for proteins involved in the interaction of *H. pylori* with the external environment and with the host (Figure 4). Indeed, genes coding for subunits of transporters, outer membrane proteins and LPS modifiers belong to the HrcA regulon (Figure 4, Table 1 and Appendix A). In addition, the *babA* gene, encoding a major player in the adhesion process of *H. pylori* to highly glycosylated mucins present in the human stomach [26], is down-regulated upon *hrcA* deletion (Figure 1, Table 1 and Figure 3C). This finding adds further complexity to the role played by the regulator HrcA in *H. pylori* gene regulatory network and encourages further studies to better characterize its involvement in the process of host-pathogen interaction.

Although *hrcA* is not essential, this study highlights the importance of this repressor for *H. pylori* virulence, survival and persistence inside the human stomach. With respect to that and considering the major medical threat of increasing *H. pylori* antibiotic resistance, HrcA may be regarded as a promising target for the design of new antimicrobial strategies.

## Figures and Tables

**Figure 1 microorganisms-07-00436-f001:**
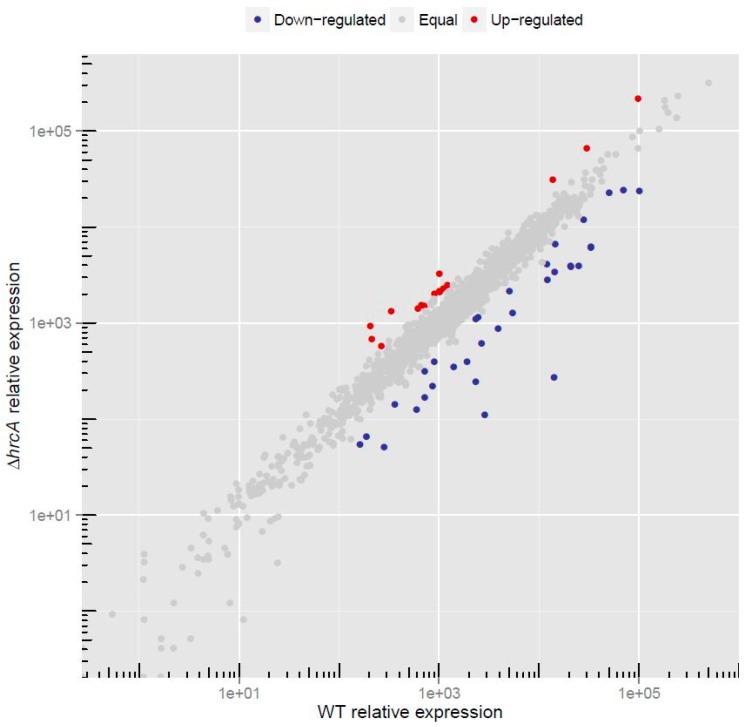
Genome-wide HrcA-dependent gene expression. The plot shows log10 scale means of the normalized counts obtained from *H. pylori* G27 wild type (x axis) and from the isogenic Δ*hrcA* deletion mutant (y axis) duplicates for each of the expressed genes. Genes found to be up-regulated, down-regulated or unchanged comparing Δ*hrcA* and wild type in DESeq2 analysis are represented as red, blue and grey dots, respectively.

**Figure 2 microorganisms-07-00436-f002:**
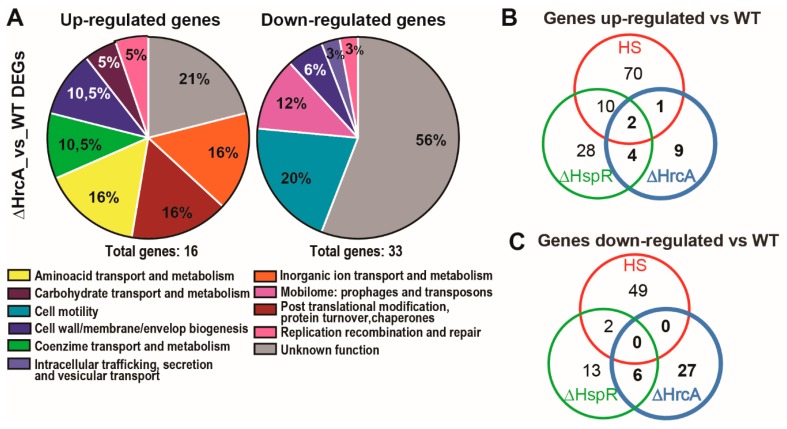
Analysis of the HrcA regulon and its integration with HspR-mediated regulation and general heat-shock response. (**A**): Pie charts showing COGs functional annotation of the differentially expressed genes outlined in the *ΔhrcA*_vs_WT comparison subdivided into up-regulated (left) and down-regulated (right) groups. The abundance of each category is indicated as a percentage, while the total number of up- and down-regulated genes is reported below each chart. (**B**,**C**): Venn diagrams showing the number of coherently up-regulated (**B**) or down-regulated (**C**) genes specific of the HrcA regulon or in common with the HspR and/or the heat-shock dataset (according to [9]).

**Figure 3 microorganisms-07-00436-f003:**
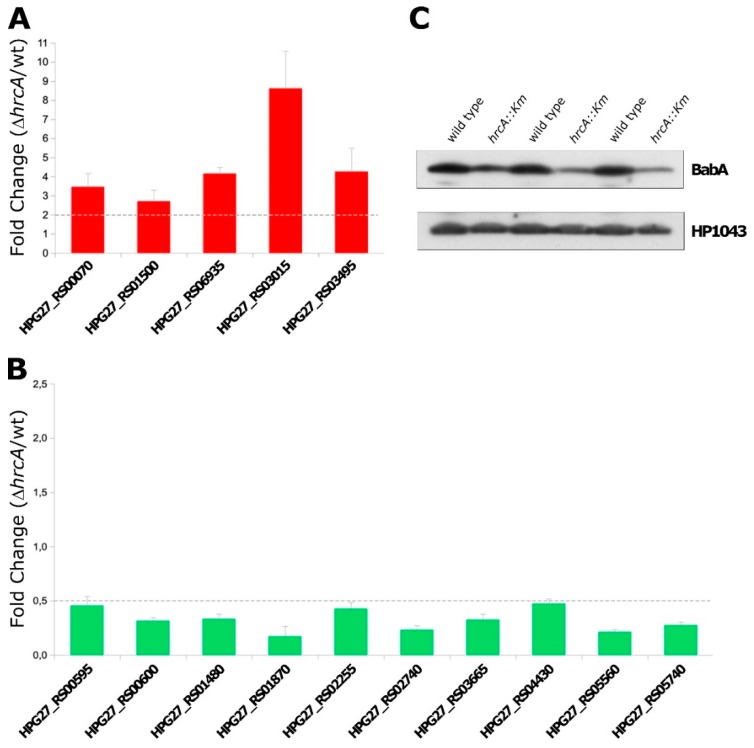
RNA-sequencing data validation. (**A**,**B**): Real time (qRT-PCR) analysis of a selection of up- (panel A) or down-regulated (panel B) genes upon *hrcA* gene deletion. Total RNA was extracted from *H. pylori* G27 wild type and *hrcA*-mutant cells and reverse transcribed to cDNA. Transcript levels of genes were quantified by qRT-PCR, using the housekeeping 16S rRNA gene as control. Error bars indicate the standard deviation deriving from three independent biological samples, each analysed in duplicate technical replicates. (**C**): Immunoblot analysis of total protein extracts of *H. pylori* G27 wild type and *hrcA*-mutant cells stained with α-BabA and α-HP1043 antibodies [18]. Immunoblot analysis was carried out on three different *H. pylori* G27 wild type and *ΔhrcA* biological replicates.

**Figure 4 microorganisms-07-00436-f004:**
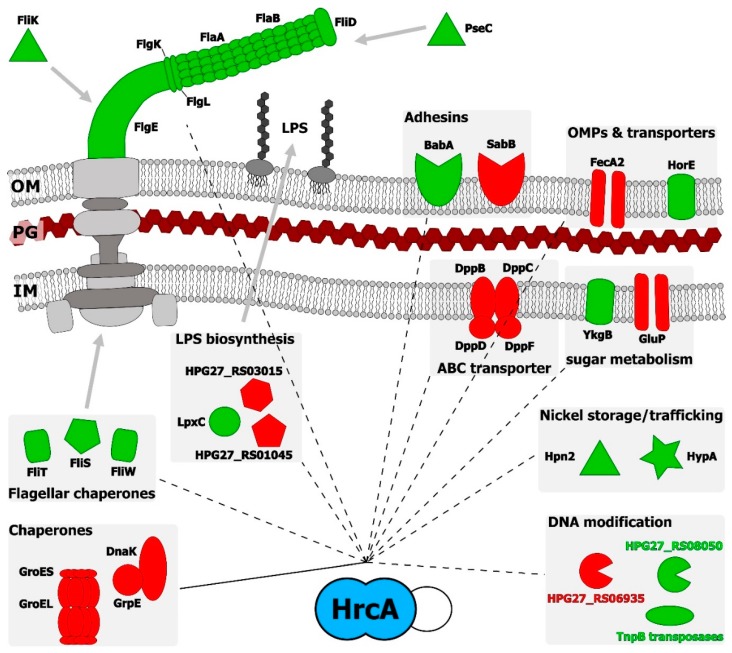
The network connects HrcA to the regulated target proteins. Filled lines indicate verified direct regulation, while dashed lines depict uncertain direct or indirect connections (not experimentally verified). HrcA-mediated positive or negative effects on genes’ expression are represented in green and red, respectively. HPG27_RS03705, HPG27_RS04725 and genes coding for hypothetical protein with unknown function have not been included in the network representation. Symbols are as follows: IM: Inner membrane; PG: Peptidoglycan layer; outer membrane; LPS: Lipopolysaccharide.

**Table 1 microorganisms-07-00436-t001:** List of up- and down-regulated genes.

Gene Names	Log2 FC	Common Names	Description
HPG27_RS00240	−2.4	*tnpB*	transposase
HPG27_RS00595	−2.2	*HPG27_RS00595*	motility accessory factor
HPG27_RS00600	−2.1	*flaB*	flagellin B
HPG27_RS01480	−2.1	*flgL*	flagellar hook-associated protein FlgL
HPG27_RS01595	−1.2	*babA, omp28*	membrane protein (adhesin)
HPG27_RS01870	−1.9	*HPG27_RS01870*	hypothetical protein
HPG27_RS00070	1.1	*groEL, hspB, hsp60*	molecular chaperone GroEL
HPG27_RS00075	1.1	*groES, hspA, hsp10*	co-chaperone GroES
HPG27_RS00625	1.0	*HPG27_RS00625*	hypothetical protein
HPG27_RS01045	1.0	*HPG27_RS01045*	LPS biosynthesis protein
HPG27_RS01500	1.0	*dppB*	peptide ABC transporter permease
HPG27_RS01505	1.2	*dppC*	peptide ABC transporter
HPG27_RS01510	1.1	*dppD*	ABC transporter ATP-binding protein
HPG27_RS01515	1.1	*dppF*	ABC transporter ATP-binding protein
HPG27_RS03015	2.0	*HPG27_RS03015*	LPS biosynthesis protein
HPG27_RS03495	1.2	*hopO, omp16, sabB*	Membrane protein (adhesin)
HPG27_RS03705	1.0	*HPG27_RS03705*	5-formyltetrahydrofolate cyclo-ligase
HPG27_RS03940	1.2	*fecA2, fecA, fecA_2*	ligand-gated channel
HPG27_RS05840	1.0	*gluP*	glucose/galactose MFS transporter
HPG27_RS06930	1.6	*HPG27_RS06930*	hypothetical protein
HPG27_RS06935	1.9	*HPG27_RS06935*	restriction endonuclease
HPG27_RS08000	1.7	*HPG27_RS08000*	hypothetical protein
HPG27_RS01870	−1.9	*HPG27_RS01870*	hypothetical protein
HPG27_RS01990	−1.1	*lpxC, envA*	UDP-3-O-[3-hydroxymyristoyl] N-acetylglucosamine deacetylase
HPG27_RS02255	−2.0	*horE, omp11*	membrane protein
HPG27_RS02575	−2.4	*HPG27_RS02575*	transposase
HPG27_RS02740	−3.1	*ykgB*	membrane protein
HPG27_RS02925	−1.5	*flaA*	flagellin A
HPG27_RS03550	−2.4	*tnpB*	transposase
HPG27_RS03660	−1.2	*fliD*	flagellar filament capping protein FliD
HPG27_RS03665	−1.1	*fliS*	flagellar chaperone protein FliS
HPG27_RS03670	−1.2	*fliT*	flagellar chaperone protein FliT
HPG27_RS04250	−1.1	*hypA*	hydrogenase/urease Ni incorporation protein HypA
HPG27_RS04255	−2.1	*flgE*	flagellar hook protein FlgE
HPG27_RS04430	−1.9	*fliK*	flagellar hook-length control protein FliK
HPG27_RS04645	−2.4	*tnpB*	transposase
HPG27_RS04725	−4.5	*HPG27_RS04725*	ATPase
HPG27_RS04730	−3.2	*HPG27_RS04730*	hypothetical protein
HPG27_RS04735	−2.6	*tnpB*	transposase
HPG27_RS04920	−1.2	*HPG27_RS04920*	hypothetical protein
HPG27_RS05380	−1.1	*pseC*	UDP-4-amino-4,6-dideoxy-N-acetyl-β-L-altrosamine transaminase (flagellar modification)
HPG27_RS05560	−2.1	*flgK*	flagellar hook-associated protein FlgK
HPG27_RS05565	−2.1	*HPG27_RS05565*	hypothetical protein
HPG27_RS05740	−1.1	*fliW*	flagellar assembly protein FliW
HPG27_RS06175	−1.8	*HPG27_RS06175*	hypothetical protein
HPG27_RS07055	−1.1	*HPG27_RS07055*	hypothetical protein
HPG27_RS07080	−1.5	*HPG27_RS07080*	nickel transporter
HPG27_RS08050	−2.0	*HPG27_RS08050*	exonuclease VII large subunit
HPG27_RS08160	−2.0	*HPG27_RS08160*	hypothetical protein
HPG27_RS08320	−1.2	*HPG27_RS08320*	hypothetical protein

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
