# Peer review of "Helicobacter pylori* Stress-Response: Definition of the HrcA Regulon"

_microorganisms, 2019, doi:10.3390/microorganisms7100436_

Round 1
Reviewer 1 Report
Dear Authors,
I have reviewed the manuscript entitled "Helicobacter pylori stress-response: definition of the HrcA regulon" and found the research well presented and of interest for the readers of this journal, therefore I recommend this communication to be accepted in present form.
Author Response
This reviewer required no revisions.
Reviewer 2 Report
Authors present a comprehensive transcriptomic analysis for identification of the HrcA regulon in the pathogenic bacterium Helicobacter pylori. In this study the wild type and the mutant, which has not transcriptional regulator HrcA, were used for comparative transcriptome analysis. Through the use of the hrcA-mutant strain authors revealed that this heat-shock repressor is involved in the regulation of 49 genes that encode different proteins. These proteins are involved in diverse cellular processes. The importance of this transcriptional repressor for H. pylori virulence, survival and persistence inside of the human body was shown. Authors suggest that HrcA may be considered as a possible “target for the design of new antimicrobial strategies”.
The text of the manuscript is well written and accompanied by clear diagrams, tables and figures.
There are some minor remarks.
1) For how many hours bacterial strains were growing in liquid and on solid medium?
Please, indicate this information in the section “2.1 Bacterial strains and growth conditions”.
2) What about a growth rate difference (if any) between these two strains (the wild type and the mutant)?
3) The Statistics section has to be included in the “Materials and Methods”.
4) It may be useful to stress shortly the scientific and practical importance of the study in the Abstract.
Author Response
Please find below the point-by-point response to the comments:
Point1:
Growing conditions wild type and HrcA mutant strains have been added on page 2, lines 73-77.
Point 2:
Differences between wild type and HrcA mutant growth rates have been added on page 2, lines 77-79
Point 3:
The required statistics method was already embedded throughout the text in the materials and method section on page 2, paragraph 2.3 RNA-sequencing: library preparation, sequencing and analysis.
Point 4:
We stressed the scientific and practical importance of the study in the Abstract on page 1, lines 28-30